# Evaluation of Blood-Based Diagnostic Biomarkers for Canine Cognitive Dysfunction Syndrome

**DOI:** 10.3390/ani15131974

**Published:** 2025-07-04

**Authors:** Jun-Won Yoon, Chan-Sik Nam, Kwang-Sup Lee, Tae-Jung Dan, Hee-Jung Jeon, Mi-Ae Kang, Hee-Myung Park

**Affiliations:** 1Department of Veterinary Internal Medicine, College of Veterinary Medicine, Konkuk University, Seoul 05029, Republic of Korea; dkrk9454@gmail.com (J.-W.Y.); enzmahsnam@naver.com (C.-S.N.); deu02181@naver.com (K.-S.L.); dantae162534@naver.com (T.-J.D.); 2VET and GENE, Seongnam 13209, Republic of Korea; hj.jeon@vetngene.co.kr (H.-J.J.); ma.kang@vetngene.co.kr (M.-A.K.)

**Keywords:** canine cognitive dysfunction syndrome, neurodegeneration, behavioral questionnaire, amyloid-beta, neurofilament light chain, glial fibrillary acidic chain

## Abstract

As dogs grow older, some begin to show changes in memory, behavior, and awareness, similar to early signs of Alzheimer’s disease in people. This condition, called cognitive dysfunction, can affect their quality of life. In this study, we aimed to find out whether certain substances in the blood could help detect and measure cognitive decline in aging dogs. We also wanted to see if combining blood tests with behavior questionnaires could improve diagnosis. We studied 77 dogs, both healthy and affected, and measured substances related to brain health. We found that one blood marker increased in dogs with cognitive problems, while others did not show a clear link. When we combined the blood test results with owner-reported behavior changes, the ability to identify affected dogs improved. Our findings suggest that simple blood tests, used together with behavioral assessments, could help veterinarians diagnose cognitive decline more accurately and earlier. This could lead to better management and care for older dogs, helping them maintain a better quality of life as they age. Early detection can also guide owners in making informed decisions to support their dogs’ health and well-being.

## 1. Introduction

Canine cognitive dysfunction syndrome (CDS) is a progressive neurodegenerative disorder that primarily affects aging dogs, with prevalence increasing to 41% in those over 14 years old [1,2]. The syndrome is characterized by disorientation, altered social interactions, disrupted sleep–wake cycles, inappropriate elimination, anxiety, and changes in activity levels, collectively known as DISHAA [3]. Diagnosis traditionally relies on behavioral questionnaires such as the Canine Cognitive Dysfunction Rating Scale (CCDR), Canine Dementia Scale (CADES), and Canine Cognitive Assessment Scale (CCAS), which, while useful, are subjective and lack definitive biological markers [4,5,6].

Given the similarities between canine cognitive dysfunction syndrome (CDS) and Alzheimer’s disease (AD) in humans, particularly in amyloid precursor protein (APP) processing, amyloid plaque deposition, and cognitive impairment [7], studies have investigated shared pathophysiological markers. Amyloid-beta (Aβ), a peptide derived from APP, is a key component of plaque formation in neurodegenerative diseases such as AD.In AD, amyloid-beta 42 (Aβ42) accumulates in the brain while its blood levels decrease, whereas amyloid-beta 40 (Aβ40) remains relatively stable [8]. Neurofilament light chain (NfL), a structural protein released during axonal damage, is elevated in plasma and correlates with disease severity [9]. Glial fibrillary acidic protein (GFAP), an astrocytic marker, also increases in AD, reflecting neuroinflammation and gliosis [10].

In veterinary medicine, the role of blood biomarkers in CDS diagnosis remains underexplored. Regarding amyloid-beta, studies have shown that Aβ42 accumulation in the brain precedes Aβ40 and is more strongly associated with total amyloid burden. In cerebrospinal fluid (CSF), Aβ42 decreases with age while Aβ40 remains stable, making the Aβ42/40 ratio a useful marker of cerebral amyloid deposition [11]. Additionally, CSF Aβ42 levels are positively associated with body weight and negatively associated with age, though not correlated with cognitive performance [12]. Studies assessing blood Aβ40 and Aβ42 in canine CDS have reported considerable variability across findings [13,14,15].

NfL has been widely studied in various canine neurologic diseases. It has shown utility as a marker of neuroaxonal injury in meningoencephalomyelitis of unknown etiology (MUE) [16]. Serum NfL levels are also significantly higher in structural epilepsy than in idiopathic epilepsy [17] and have been used to distinguish structural neurologic conditions, including MUE and hydrocephalus [18]. In CDS, NfL has consistently been elevated in affected dogs compared to cognitively normal individuals, suggesting its relevance as a diagnostic marker in age-related neurodegeneration [19,20].

GFAP has been shown to play an important role in various central nervous system diseases in dogs. In necrotizing meningoencephalitis (NME), GFAP levels are significantly elevated, and CSF concentrations of GFAP have been found to positively correlate with anti-GFAP IgG titers in affected dogs, whereas such associations are not observed in dogs with other neurologic conditions [21]. Moreover, the prognostic utility of serum GFAP has been demonstrated in cases of progressive myelomalacia, in which serum levels exhibited a strong correlation with clinical disease progression and severity [22]. These findings suggest that GFAP may reflect underlying astrocytic damage and inflammation in a range of neurodegenerative or inflammatory diseases. However, despite its demonstrated relevance in other neurologic conditions, the potential role of GFAP in CDS has not yet been specifically investigated or validated.

This study aimed to assess the diagnostic value of selected blood-based biomarkers, including Aβ40, Aβ42, NfL, and GFAP, for identifying and staging canine cognitive dysfunction syndrome (CDS) in aging dogs. We also examined whether combining these objective biomarkers with standardized behavioral questionnaires could enhance the diagnostic accuracy, consistency, and clinical utility of CDS evaluation in veterinary practice. To our knowledge, this is the first study to simultaneously apply all three widely validated CDS questionnaires—CCDR, CADES, and CCAS—within a single cohort, enabling a comprehensive and comparative analysis of cognitive staging tools. Drawing on previous findings from veterinary neurodegenerative research, we hypothesized that blood Aβ40 and Aβ42 concentrations would decrease with disease progression, while NfL and GFAP levels would increase, reflecting axonal degeneration and glial activation, respectively.

## 2. Materials and Methods

### 2.1. Subjects

This study enrolled a total of 77 dogs, comprising both healthy dogs and those showing signs of CDS. Recruitment was conducted at six local animal hospitals. Dogs were categorized based on age and cognitive status using three different questionnaires (CCDR, CADES, and CCAS; see Appendix A). All dogs were included in this study, and each dog was independently classified according to the criteria of each questionnaire. A young control group consisting of 14 clinically healthy dogs under 7 years of age, with no signs of CDS, was also included. The remaining dogs, aged 7 years or older, were divided into three or four groups based on their cognitive status as separately assessed by each questionnaire. Breed information was obtained through owner-completed questionnaires. The sample collection period extended from June 2024 to February 2025.

### 2.2. Sample Collection

Blood samples were collected from all dogs to measure concentrations of Aβ40, Aβ42, NfL, and GFAP. Fasting venous blood was drawn from the jugular or cephalic vein. Plasma samples were collected into K2 EDTA tubes and centrifuged at 2000× *g* for 8 min; the supernatant was stored at −70 °C. Serum samples were obtained using serum separator tubes (SSTs), allowed to clot at room temperature for 30 min, and then centrifuged at 3000 rpm for 15 min. The serum was stored at −70 °C until biomarker analysis.

### 2.3. Questionnaires

To diagnose CDS, a behavioral questionnaire survey was conducted with owner consent. Three validated questionnaires widely used in previous research—CCDR, CADES, and CCAS [4,5,6]—were utilized to assess cognitive impairment in dogs. These tools are based on established CDS symptoms and are recognized for evaluating cognitive decline in aging dogs. The CCDR consists of 13 questions across four domains, with responses scored on a 1–5 scale, categorizing dogs as normal, at-risk, or CDS. The CADES includes 17 questions across four domains, scored on a 0–4 scale (excluding 1-point responses), and classifies dogs as normal, mild, moderate, or severe. The CCAS comprises 17 questions across six domains, scored on a 0–3 scale, and categorizes dogs as normal, mild, or severe.

### 2.4. Blood-Based Biomarkers

Blood-based biomarkers were measured using enzyme-linked immunosorbent assay (ELISA) according to the manufacturers’ instructions. Plasma samples were used to quantify Aβ40, Aβ42, and GFAP, while serum samples were used for NfL measurement. The concentrations of Aβ40 and Aβ42 were determined using the Canine Amyloid Beta Peptide 1-40 ELISA Kit and the Canine Amyloid Beta Peptide 1-42 ELISA Kit (MyBiosource, San Diego, CA, USA), respectively. GFAP concentrations were measured with the Human GFAP ELISA Kit (Biovendor Laboratory Medicine, Modrice, Czech Republic). Serum NfL concentrations were assessed using the NF-light ELISA Kit (Umandiagnostics, Umeå, Sweden). ELISA measurements were performed using the SpectraMax ABS Plus microplate reader (Molecular Devices, San Jose, CA, USA) at 450 nm with 630 nm as the reference wavelength. The assays showed high accuracy determined by linearity under dilution (coefficient of correlation close to 1). All samples were run in duplicate, and mean values were used for analysis to ensure precision and reliability; final concentrations were calculated based on the respective standard curves.

### 2.5. Data Analysis

Statistical analyses were performed using IBM SPSS Statistics version 30.0 (IBM Corp., Armonk, NY, USA). Since the data violated normality assumptions, non-parametric tests were applied. For each biomarker, median and interquartile range (IQR; Q1–Q3) were calculated. Group differences in plasma or serum parameters were assessed by ANOVA (parametric) or Kruskal–Wallis test (non-parametric), followed by Dunn’s post hoc test with Bonferroni correction. Receiver operating characteristic (ROC) curve analysis was used to evaluate diagnostic performance by calculating the area under the curve (AUC). Data are presented as median (Q1–Q3) in box plots; whiskers represent values within 1.5× IQR. Statistical significance for all analyses was defined as *p* < 0.05.

## 3. Results

### 3.1. Population and Ages

The distribution of the 77 dogs included in this study was analyzed by sex, breed, neuter status, and age. Fourteen dogs under 7 years of age were classified as young controls (mean age 3.20 ± 1.90 years), showing no clinical signs of cognitive decline. The remaining 63 dogs, aged 7 years or older, were categorized into cognitive stages based on behavioral questionnaire scores. Using the CCDR, 32 dogs were classified as aged controls, 11 as at-risk, and 20 as CDS. Based on the CADES, 8 were aged controls, 23 were mild, 17 were moderate, and 15 were severe. According to the CCAS, 14 dogs were aged controls, 32 were mild, and 17 were severe. These classifications served as the basis for group comparisons and biomarker analyses. Detailed group distributions and mean ages are shown in Table 1, Table 2 and Table 3.

Age was positively associated with cognitive decline across all three assessment tools. Dogs with more severe impairment were generally older. In CCDR-based groups, age significantly increased from aged controls to CDS cases (Figure 1A). Similarly, CADES-classified moderate and severe dogs were older than those with milder scores (Figure 1B), and CCAS groups showed a progressive age increase from controls to mild and severe stages (Figure 1C).

No statistically significant differences were found in sex distribution among the cognitive groups. Neutered dogs were more frequently represented than intact dogs across the entire cohort. In terms of breed representation, Maltese, Poodles, and mixed breeds were the most common breeds across all three assessment tools.

### 3.2. Concentrations of Blood-Based Biomarkers

Based on the classifications obtained from the three CDS questionnaires, dogs were grouped according to their cognitive status. The concentrations of Aβ40, Aβ42, NfL, and GFAP were subsequently measured and compared across these groups to assess their association with disease severity. The detailed biomarker concentrations for each group are presented in Table 4.

#### 3.2.1. Amyloid-Beta 40,42 Concentrations

The young control (YC) group showed a median (Q1–Q3) Aβ-40 concentration of 34.35 (28.93–37.64) pg/mL, Aβ-42 of 22.20 (18.86–29.13) pg/mL, and an Aβ-42:Aβ-40 ratio of 0.66 (0.61–0.74), reflecting baseline levels in cognitively normal dogs. Under the CCDR classification, Aβ-40 concentrations were 32.57 (29.73–35.13) pg/mL in the aged control (AC) group, 34.19 (30.47–34.69) in the at-risk group, and 35.31 (28.19–38.77) in the CDS group. Aβ-42 levels were measured at 21.36 (20.04–25.47), 22.51 (17.92–24.77), and 21.97 (19.04–26.35) pg/mL in the AC, at-risk, and CDS groups, respectively. The corresponding Aβ-42:Aβ-40 ratio was 0.67 (0.62–0.75) in the AC group, 0.71 (0.64–0.75) in the at-risk group, and 0.68 (0.58–0.71) in the CDS group, showing only subtle variation. According to the CADES classification, Aβ-40 concentrations were 33.87 (32.32–36.58) pg/mL in the AC group, 30.29 (27.32–32.89) in the mild group, 34.67 (30.68–36.05) in the moderate group, and 34.51 (30.04–38.63) in the severe group. Aβ-42 levels ranged from 21.33 (19.37–27.07) in the AC group to 22.13 (18.23–26.54) in the severe group, with overlapping interquartile ranges across stages. The Aβ-42:Aβ-40 ratio varied from 0.62 (0.62–0.70) in the AC group to 0.70 (0.65–0.77) in the mild group, 0.67 (0.63–0.72) in the moderate group, and 0.67 (0.58–0.71) in the severe group. Under the CCAS classification, Aβ-40 concentrations were 32.99 (30.43–35.62) pg/mL in the AC group, 32.08 (28.90–35.57) in the mild group, and 34.51 (30.87–38.29) in the severe group. Aβ-42 levels ranged from 21.04 (18.85–25.38) to 22.85 (18.90–27.05) pg/mL, showing slight overlap across stages. The Aβ-42:Aβ-40 ratio was 0.63 (0.61–0.74) in the AC group, 0.70 (0.65–0.75) in the mild group, and 0.65 (0.59–0.71) in the severe group.

Taken together, plasma Aβ-40 concentrations (Figure 2), Aβ-42 concentrations (Figure 3), and the Aβ-42 to Aβ-40 ratio (Figure 4) did not significantly differ across any stage of CDS, regardless of the classification method used.

#### 3.2.2. Neurofilament Light Chain Concentration

The young control (YC) group exhibited a serum NfL concentration of 8.59 (5.13–20.03) pg/mL, indicating low baseline levels in cognitively healthy dogs. Under the CCDR classification, NfL concentrations increased to 14.57 (3.72–26.35) pg/mL in the aged control (AC) group, 43.24 (11.49–64.85) in the at-risk group, and 50.30 (20.62–63.40) in the CDS group. Statistically significant differences were observed between the CDS group and both control groups (*p* < 0.05), reflecting the progressive elevation of NfL with increasing cognitive impairment (Figure 5A). According to the CADES classification, NfL levels also showed a clear upward trend: 3.38 (2.52–18.61) pg/mL in the AC group, 14.57 (4.70–31.52) in the mild group, 43.24 (14.11–68.46) in the moderate group, and 52.28 (21.12–63.67) in the severe group. Statistically significant elevations were noted in the moderate and severe groups compared to controls (*p* < 0.05) (Figure 5B). This stepwise increase further supports the correlation between disease severity and NfL concentration. Within the CCAS system, NfL concentrations were 3.38 (2.47–16.04) pg/mL in the AC group, increasing to 22.33 (10.34–60.81) in the mild group and 45.78 (17.86–61.77) in the severe group. Dogs with mild and severe impairment had significantly higher NfL levels than the AC group, and a marginally significant increase was also observed between the YC and severe groups (*p* = 0.058), suggesting a potential early elevation of serum NfL in the transition from normal aging to advanced cognitive decline (Figure 5C).

These findings consistently demonstrate that serum NfL concentrations increase in accordance with cognitive impairment severity across all classification systems. Taken together, they support the value of NfL as a sensitive and reliable peripheral biomarker for detecting neurodegeneration and staging CDS in aging dogs—even in its preclinical or subthreshold stages.

#### 3.2.3. Glial Fibrillary Acidic Protein Concentration

The young control (YC) group showed a plasma GFAP concentration of 0.286 (0.270–0.313) ng/mL, representing the baseline level in neurologically healthy dogs. In the CCDR classification, GFAP levels were 0.270 (0.253–0.281) ng/mL in the aged control (AC) group, 0.203 (0.176–0.284) ng/mL in the at-risk group, and 0.238 (0.210–0.268) ng/mL in the CDS group, with a slight decrease observed in the at-risk and CDS stages compared to AC. According to the CADES classification, GFAP values were 0.281 (0.271–0.293) ng/mL in AC dogs, followed by 0.269 (0.253–0.285) in the mild group, 0.222 (0.202–0.254) in the moderate group, and 0.237 (0.200–0.264) in the severe group, reflecting a subtle downward trend with increasing severity. In the CCAS classification, GFAP concentrations were recorded as 0.276 (0.271–0.292) ng/mL in the AC group, 0.253 (0.206–0.268) in the mild group, and 0.237 (0.211–0.283) in the severe group, again suggesting a modest reduction in GFAP levels as cognitive dysfunction progressed.

Overall, although minor group-wise differences and decreasing tendencies were observed, plasma GFAP levels did not show any statistically significant variation in relation to CDS severity across all three classification systems (Figure 6).

### 3.3. ROC Curve Analysis of Serum NfL Concentration

Given the significantly elevated serum NfL concentrations observed in dogs diagnosed with CDS, a receiver operating characteristic (ROC) curve analysis was performed to further evaluate the diagnostic performance of serum NfL in distinguishing CDS-affected dogs from cognitively normal controls.

When classified according to the CCDR, the area under the curve (AUC) was 0.763 (95% confidence interval [CI]: 0.649–0.877, *p* < 0.001), indicating a good level of diagnostic accuracy. The optimal cut-off value for differentiating CDS-affected dogs from aged controls was calculated to be 18.28 pg/mL (Figure 7A). Using the CADES classification, the AUC was determined to be 0.722 (95% CI: 0.599–0.845, *p* < 0.001), reflecting moderate discriminatory ability, with a higher optimal cut-off value of 43.13 pg/mL identified (Figure 7B). In the CCAS-based classification, ROC analysis yielded an AUC of 0.777 (95% CI: 0.664–0.889, *p* < 0.001), which also suggested good diagnostic performance. Interestingly, the same cut-off value of 43.13 pg/mL was identified as optimal for this classification system as well (Figure 7C).

## 4. Discussion

This study investigated the relationship between CDS severity and blood biomarkers, including Aβ40, Aβ42, NfL, and GFAP, and evaluated the diagnostic utility of owner-based behavioral questionnaires. A significant association between CDS stage and age was identified, with older dogs showing more severe cognitive impairment. These findings align with previous reports highlighting age-related progression of CDS [23]. However, variability in classification criteria across questionnaires suggests that establishing a standardized diagnostic framework would enhance consistency and clinical applicability.

In line with previous studies, sex and neuter status were not significantly associated with the presence or severity of CDS in this cohort [24]. Regarding breed distribution, Maltese, Poodles, and mixed-breed dogs were the most represented, reflecting the breed demographics typical of companion dogs in South Korea [24].

Regarding biomarkers, previous studies have suggested that the plasma Aβ42/40 ratio may serve as an early indicator of CDS progression [13,14,15]. Some studies reported higher plasma Aβ concentrations in younger dogs compared to aged cognitively normal or CDS-affected dogs [14], reflecting peripheral declines in Aβ as brain deposition increases—a process that has been well-documented in both dogs and humans, particularly in the context of aging [25]. However, in this study, plasma Aβ40, Aβ42, and Aβ42/40 levels did not significantly differ between groups. This inconsistency may stem from individual and breed-specific differences in amyloid dynamics, plateau effects in amyloid accumulation, or confounding by mixed pathologies. Several human studies suggest that amyloid accumulation occurs prior to the onset of clinical symptoms and tends to plateau, showing minimal or no further increase even as the disease continues to clinically progress [14]. These findings suggest that while brain amyloid pathology may exist, peripheral Aβ measurements in dogs may not reliably reflect disease progression, contrasting with trends observed in human Alzheimer’s disease.

Among the biomarkers analyzed, serum NfL demonstrated the strongest and most consistent association with CDS severity across all three behavioral scales. Dogs classified as CDS by CCDR, severe by CADES, and mild to severe by CCAS showed significantly elevated serum NfL levels. These results are consistent with previous studies demonstrating increased plasma NfL concentrations in dogs with mild cognitive impairment [20], positive correlations between plasma NfL and CADES scores independent of weight and sex [19], and elevated NfL levels detected via immunomagnetic reduction assays [26]. The biological rationale for these findings lies in the role of NfL in maintaining axonal structure and function, with neuroaxonal damage leading to NfL release into cerebrospinal fluid and blood [19].

ROC curve analyses reinforced the clinical relevance of serum NFL, yielding AUCs of 0.763, 0.722, and 0.777 for CCDR, CADES, and CCAS, respectively, indicating moderate to good diagnostic performance. Interestingly, the derived cut-off values varied across the tools: 18.28 pg/mL for CCDR and 43.13 pg/mL for both CADES and CCAS. These discrepancies may reflect intrinsic differences in scale sensitivity and diagnostic thresholds; CCDR may capture cognitive impairment at a lower NFL concentration, whereas CADES and CCAS might require higher thresholds due to stricter scoring algorithms or differences in item structure. The identical cut-off for CADES and CCAS likely reflects their structural similarities, consistent with previously reported high correlations between their scores [27]. These findings highlight the importance of considering the characteristics of each assessment tool when interpreting biomarker thresholds and suggest that while NFL shows consistent trends across scales, its diagnostic cut-off may need to be tailored to the specific cognitive instrument used.

In contrast, GFAP did not show statistically significant differences across CDS stages, despite a decreasing trend. This diverges from human studies where plasma GFAP increases in neurodegenerative conditions [10,28]. In dogs, the presence of circulating anti-GFAP autoantibodies has been reported [21,29,30], potentially interfering with GFAP measurement and complicating its interpretation as a biomarker for CDS. Further studies are warranted to explore the prevalence of these autoantibodies and to refine GFAP quantification methods in canine neurodegeneration.

Several limitations should be acknowledged. The definition of “senior” dogs is also clinically relevant. Although a 7-year threshold is commonly used [31], particularly in medium- to large-breed dogs such as Beagles and Border Collies, its applicability may be limited in regions like South Korea, where small breeds predominate and often live beyond 15 years. A later cutoff, such as 9 or 10 years, may more accurately reflect aging patterns in these populations. Further breed- and region-specific studies are needed to refine age classifications for early CDS detection and staging. The absence of magnetic resonance imaging (MRI) also precluded anatomical confirmation of brain changes and exclusion of other neuropathologies. While practical, ELISA may lack sensitivity for low-abundance proteins compared to newer technologies such as Single Molecule Array (SIMOA). Biomarker levels were measured in blood rather than cerebrospinal fluid, potentially underestimating central nervous system pathology. Lastly, cognitive staging relied on owner-based questionnaires, which, despite validation, introduce subjectivity. Incorporating imaging and standardized cognitive testing could enhance diagnostic precision in future studies. We further recognize that single-timepoint blood sampling may inadequately capture the progressive and variable nature of CDS, potentially limiting the interpretability of our findings. Future studies will incorporate longitudinal sampling to improve biomarker validation and more precisely delineate individual cognitive trajectories. Despite these limitations, our findings support serum NfL as a promising blood-based biomarker for diagnosing and staging CDS in aging dogs, with the potential to complement behavioral assessments and enhance early detection and clinical management of canine cognitive decline.

## 5. Conclusions

This study highlights the potential of serum NfL as a reliable blood-based biomarker for diagnosing and staging CDS. NfL levels showed a clear association with disease severity, suggesting its clinical utility in detecting neurodegeneration. While Aβ40, Aβ42, and GFAP were not significantly correlated with CDS stage, their roles in disease progression cannot be fully excluded and warrant further investigation.

Importantly, combining questionnaire-based assessments with serum NfL measurements may enhance diagnostic accuracy and objectivity, particularly in settings where advanced imaging is not available. These findings support the integration of blood biomarkers and behavioral assessments into routine veterinary practice for improved CDS evaluation.

## Figures and Tables

**Figure 1 animals-15-01974-f001:**
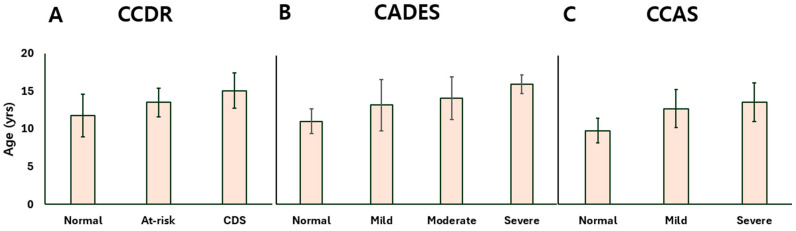
Bar plots presenting the age distribution of dogs at different stages of CDS as determined by the (**A**) CCDR, (**B**) CADES, and (**C**) CCAS questionnaires. CDS: canine cognitive dysfunction syndrome; CCDR: Canine Cognitive Dysfunction Rating Scale; CADES: Canine Dementia Scale; CCAS: Canine Cognitive Assessment Scale.

**Figure 2 animals-15-01974-f002:**
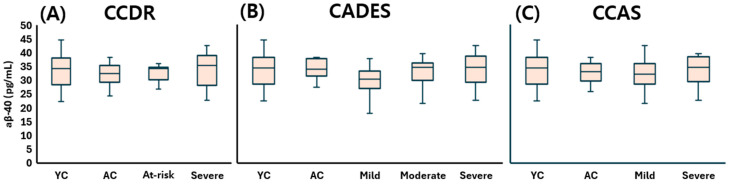
Comparison of plasma Aβ-40 levels across cognitive stages in dogs, classified by three cognitive assessment tools: (**A**) CCDR, (**B**) CADES, and (**C**) CCAS. Box plots illustrate group distributions based on the CCDR, CADES, and CCAS. Groups include YC, AC, and dogs with varying degrees of cognitive impairment. Medians are shown as horizontal lines; boxes represent interquartile ranges, and whiskers extend to values within 1.5× the IQR. Aβ-40: amyloid beta 40; CCDR: Canine Cognitive Dysfunction Rating Scale; CADES: Canine Dementia Scale; CCAS: Canine Cognitive Assessment Scale; YC: young control; AC: aged control.

**Figure 3 animals-15-01974-f003:**
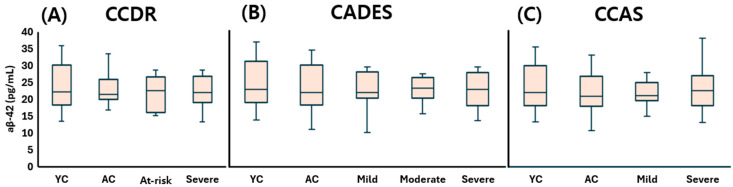
Comparison of plasma Aβ-42 levels across cognitive stages in dogs, classified by three cognitive assessment tools: (**A**) CCDR, (**B**) CADES, and (**C**) CCAS. Box plots illustrate group distributions based on the CCDR, CADES, and CCAS. Groups include YC, AC, and dogs with varying degrees of cognitive impairment. Medians are shown as horizontal lines; boxes represent interquartile ranges, and whiskers extend to values within 1.5× the IQR. Aβ-42: amyloid beta 42; CCDR: Canine Cognitive Dysfunction Rating Scale; CADES: Canine Dementia Scale; CCAS: Canine Cognitive Assessment Scale; YC: young control; AC: aged control.

**Figure 4 animals-15-01974-f004:**
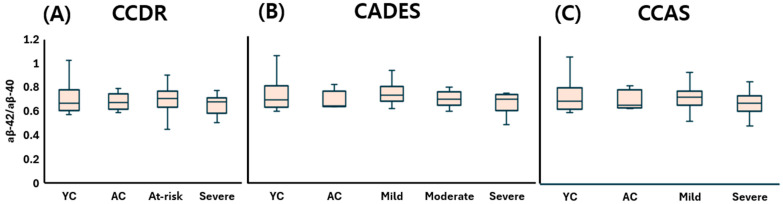
Comparison of plasma Aβ42/Aβ40 ratios across cognitive stages in dogs, classified by three different cognitive assessment tools: (**A**) CCDR, (**B**) CADES, and (**C**) CCAS. Box plots illustrate group distributions based on the CCDR, CADES, and CCAS. Groups include YC, AC, and dogs with varying degrees of cognitive impairment. Medians are shown as horizontal lines; boxes represent interquartile ranges, and whiskers extend to values within 1.5× the IQR. Aβ-40: amyloid beta 40; Aβ-42: amyloid beta 42; CCDR: Canine Cognitive Dysfunction Rating Scale; CADES: Canine Dementia Scale; CCAS: Canine Cognitive Assessment Scale; YC: young control; AC: aged control.

**Figure 5 animals-15-01974-f005:**
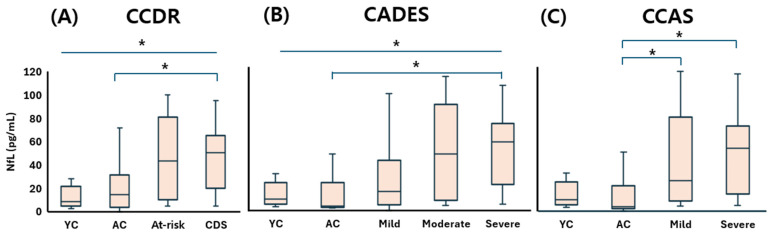
Comparison of serum NfL concentrations across cognitive stages in dogs, classified by three cognitive assessment tools: (**A**) CCDR, (**B**) CADES, and (**C**) CCAS. Box plots illustrate group distributions based on the CCDR, CADES, and CCAS. Groups include YC, AC, and dogs with varying degrees of cognitive impairment. Medians are shown as horizontal lines; boxes represent interquartile ranges, and whiskers extend to values within 1.5× the IQR. *p* < 0.05. An asterisk (*) indicates a statistically significant difference between groups (*p* < 0.05). NfL: neurofilament light chain; CDS: canine cognitive dysfunction syndrome; CCDR: Canine Cognitive Dysfunction Rating Scale; CADES: Canine Dementia Scale; CCAS: Canine Cognitive Assessment Scale; YC: young control; AC: aged control.

**Figure 6 animals-15-01974-f006:**
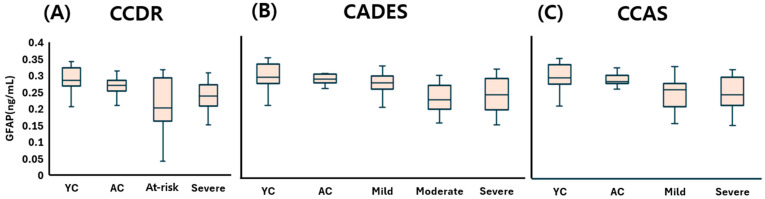
Comparison of plasma GFAP concentrations across cognitive stages in dogs, classified by three cognitive assessment tools: (**A**) CCDR, (**B**) CADES, and (**C**) CCAS. Box plots illustrate group distributions based on the CCDR, CADES, and CCAS. Groups include YC, AC, and dogs with varying degrees of cognitive impairment. Medians are shown as horizontal lines; boxes represent interquartile ranges, and whiskers extend to values within 1.5× the IQR. GFAP: glial fibrillary acidic protein; CCDR: Canine Cognitive Dysfunction Rating Scale; CADES: Canine Dementia Scale; CCAS: Canine Cognitive Assessment Scale; YC: young control; AC: aged control.

**Figure 7 animals-15-01974-f007:**
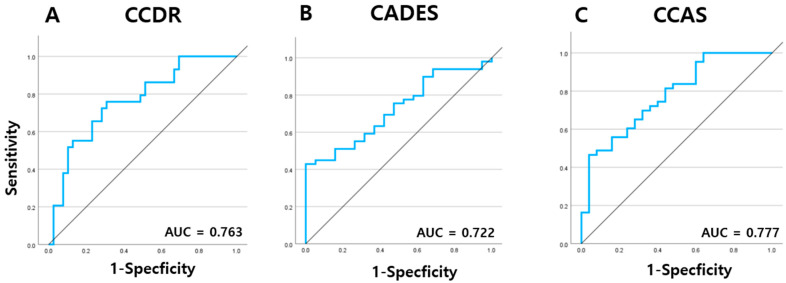
ROC curve analysis of serum NfL levels of dogs at different stages of CDS as determined by the (**A**) CCDR, (**B**) CADES, and (**C**) CCAS questionnaires. NfL: neurofilament light chain; CDS: canine cognitive dysfunction syndrome; CCDR: Canine Cognitive Dysfunction Rating Scale; CADES: Canine Dementia Scale; CCAS: Canine Cognitive Assessment Scale.

**Table 1 animals-15-01974-t001:** Characteristics of the 77 dogs in the study population as evaluated by CCDR.

Variables	<7 years	≥7 years		
	Young Control (*n* = 14)	Aged Control (*n* = 32)	At-Risk (*n* = 11)	CDS (*n* = 15)
Sex, *n* (%)				
Castrated Male	4 (28.6)	15 (46.9)	5 (45.5)	10 (50.0)
Male	1 (7.1)	1 (3.1)	1 (9.0)	0 (0.0)
Spayed Female	8 (57.1)	15 (46.9)	5 (45.5)	9 (45.0)
Female	1 (7.1)	1 (3.1)	0 (0.0)	1 (5.0)
Breed, *n* (%)				
Poodle	2 (14.3)	5 (15.6)	4 (36.4)	7 (35.0)
Maltese	0 (0.0)	7 (21.9)	4 (36.4)	3 (15.0)
Bichon Frise	1 (7.1)	2 (6.3)	0 (0.0)	2 (10.0)
Yorkshire Terrier	0 (0.0)	2 (6.3)	1 (9.0)	1 (5.0)
Shih Tzu	1 (7.1)	2 (6.3)	0 (0.0)	1 (5.0)
Mixed	2 (14.3)	6 (18.8)	1 (9.0)	5 (25.0)
Others *	8 (57.1)	8 (25.0)	1 (9.0)	1 (5.0)
Age (years)	3.20 ± 1.90	11.76 ± 2.85	13.50 ± 1.91	15.06 ± 2.36

* Others: French Bulldog, Italian Greyhound, Maltipoo, Pomeranian, Cavalier King Charles Spaniel, Labrador Retriever, Spitz, Miniature Pinscher, Long hair Chihuahua, Schnauzer.

**Table 2 animals-15-01974-t002:** Characteristics of the 77 dogs in the study population as evaluated by CADES.

Variables	<7 years	≥7 years			
	Young Control (*n* = 14)	Aged Control (*n* = 8)	Mild (*n* = 23)	Moderate (*n* = 17)	Severe (*n* = 15)
Sex, *n* (%)					
Castrated Male	4 (28.6)	4 (50.0)	12 (52.2)	5 (29.4)	9 (60.0)
Male	1 (7.1)	1 (12.5)	0 (0.0)	1 (5.9)	0 (0.0)
Spayed Female	8 (57.1)	3 (37.5)	9 (39.1)	11 (64.7)	6 (40.0)
Female	1 (7.1)	0 (0.0)	2 (8.7)	0 (0.0)	0 (0.0)
Breed, *n* (%)					
Poodle	2 (14.3)	1 (12.5)	4 (17.4)	7 (41.2)	4 (26.7)
Maltese	0 (0.0)	2 (25.0)	7 (30.4)	3 (17.6)	2 (13.3)
Bichon Frise	1 (7.1)	0 (0.0)	2 (8.7)	0 (00.0)	2 (13.3)
Yorkshire Terrier	0 (0.0)	0 (0.0)	1 (4.3)	2 (11.8)	1 (6.7)
Shih Tzu	1 (7.1)	1 (12.5)	1 (4.3)	0 (0.0)	1 (6.7)
Mixed	2 (14.3)	2 (25.0)	2 (8.7)	4 (23.5)	4 (26.7)
Others *	8 (57.1)	2 (25.0)	6 (26.1)	1 (5.9)	1 (6.7)
Age (years)	3.20 ± 1.90	10.48 ± 1.53	12.51 ± 3.21	13.36 ± 2.69	15.15 ± 1.17

* Others: French Bulldog, Italian Greyhound, Maltipoo, Pomeranian, Cavalier King Charles Spaniel, Labrador Retriever, Spitz, Miniature Pinscher, Long hair Chihuahua, Schnauzer.

**Table 3 animals-15-01974-t003:** Characteristics of the 77 dogs in the study population as evaluated by CCAS.

Variables	<7 years	≥7 years		
	Young Control (*n* = 14)	Aged Control (*n* = 14)	Mild (*n* = 32)	Severe (*n* = 17)
Sex, *n* (%)				
Castrated Male	4 (28.6)	7 (50.0)	17 (53.1)	6 (35.3)
Male	1 (7.1)	1 (7.1)	0 (0.0)	1 (5.9)
Spayed Female	8 (57.1)	5 (35.8)	14 (43.8)	10 (58.8)
Female	1 (7.1)	1 (7.1)	1 (3.1)	0 (0.0)
Breed, *n* (%)				
Poodle	2 (14.3)	2 (14.3)	8 (25.0)	6 (35.3)
Maltese	0 (0.0)	2 (14.3)	9 (28.1)	3 (17.6)
Bichon Frise	1 (7.1)	0 (0.0)	3 (9.4)	1 (5.9)
Yorkshire Terrier	0 (0.0)	1 (7.1)	2 (6.3)	1 (5.9)
Shih Tzu	1 (7.1)	1 (7.1)	2 (6.3)	0 (0.0)
Mixed	2 (14.3)	3 (21.4)	5 (15.6)	4 (23.5)
Others *	8 (57.1)	5 (35.8)	3 (9.4)	2 (11.8)
Age (years)	3.20 ± 1.90	10.42 ± 1.75	13.56 ± 2.72	14.48 ± 2.76

* Others: French Bulldog, Italian Greyhound, Maltipoo, Pomeranian, Cavalier King Charles Spaniel, Labrador Retriever, Spitz, Miniature Pinscher, Long hair Chihuahua, Schnauzer.

**Table 4 animals-15-01974-t004:** Biomarker concentrations of the 77 dogs in the study population.

Variables		*n*	Aβ-40	Aβ-42	Aβ-42/Aβ-40	NfL	GFAP
YC		14	34.35 (28.93–37.64)	22.20 (18.86–29.13)	0.66 (0.61–0.74)	8.59 (5.13–20.03)	0.286 (0.270–0.313)
CCDR	AC	32	32.57 (29.73–35.13)	21.36 (20.04–25.47)	0.67 (0.62–0.75)	14.57 (3.72–26.35)	0.270 (0.253–0.281)
	At-risk	11	34.19 (30.47–34.69)	22.51 (17.92–24.77)	0.71 (0.64–0.75)	43.24 (11.49–64.85)	0.203 (0.176–0.284)
	CDS	15	35.31 (28.19–38.77)	21.97 (19.04–26.35)	0.68 (0.58–0.71)	50.30 (20.62–63.40)	0.238 (0.210–0.268)
CADES	AC	8	33.87 (32.32–36.58)	21.33 (19.37–27.07)	0.62 (0.62–0.70)	3.38 (2.52–18.61)	0.281 (0.271–0.293)
	Mild	23	30.29 (27.32–32.89)	21.38 (19.80–25.95)	0.70 (0.65–0.77)	14.57 (4.70–31.52)	0.269 (0.253–0.285)
	Moderate	17	34.67 (30.68–36.05)	22.51 (20.10–25.29)	0.67 (0.63–0.72)	43.24 (14.11–68.46)	0.222 (0.202–0.254)
	Severe	15	34.51 (30.04–38.63)	22.13 (18.23–26.54)	0.67 (0.58–0.71)	52.28 (21.12–63.67)	0.237 (0.200–0.264)
CCAS	AC	14	32.99 (30.43–35.62)	21.04 (18.85–25.38)	0.63 (0.61–0.74)	3.38 (2.47–16.04)	0.276 (0.271–0.292)
	Mild	32	32.08 (28.90–35.57)	21.36 (20.00–24.85)	0.70 (0.65–0.75)	22.33 (10.34–60.81)	0.253 (0.206–0.268)
	Severe	17	34.51 (30.87–38.29)	22.85 (18.90–27.05)	0.65 (0.59–0.71)	45.78 (17.86–61.77)	0.237 (0.211–0.283)

Median (25th percentile–75th percentile); Aβ-40: amyloid beta 40; Aβ-42: amyloid beta 42; NfL: neurofilament light chain; GFAP: glial fibrillary acidic protein; YC: young control; AC: aged control.

## Data Availability

This article contains all data generated or analyzed during this study. Further details or datasets are available from the corresponding author upon reasonable request.

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
