# Peer review of "Evaluation of Blood-Based Diagnostic Biomarkers for Canine Cognitive Dysfunction Syndrome"

_animals, 2025, doi:10.3390/ani15131974_

Round 1
Reviewer 1 Report
Comments and Suggestions for Authors
Dear authors:
This document provides important information for the timely and simple diagnosis of cognitive disorders in dogs. This is a very useful topic today because dogs have become an important part of many families, and owners are increasingly concerned about improving their quality of life and welfare. In addition, integrating laboratory and behavioral tests helps to obtain more complete evaluations of patients. However, I have added some suggestions and comments to improve this writing.
Lines 91-100: It would be interesting if you mention the breeds and sex of the dogs that were part of the study in this paragraph, I understand that it is mentioned later in table 1, but if you could mention it briefly, or perhaps anticipate that this data will be mentioned later in said table. It would be good to generate more interest from the reader.
Lines 101-107: Please briefly mention what equipment you used to analyze the blood biomarkers.
Lines 118-140: Perhaps, instead of “blood-based biomarkers”, the title of this section could be “Statistical analysis and blood-based biomarkers”, and integrate both sections into one.
Discussion section: Please discuss why some types of breeds or sex might have shown these results more frequently than others.
Author Response
Comments 1: Lines 91-100: It would be interesting if you mention the breeds and sex of the dogs that were part of the study in this paragraph, I understand that it is mentioned later in table 1, but if you could mention it briefly, or perhaps anticipate that this data will be mentioned later in said table. It would be good to generate more interest from the reader.
Response 1: We agree. While detailed group distributions including breed and sex are provided in Tables 1–3 (as indicated in line 162), we have also added a concise summary of sex, neuter status and breed distribution in lines 183–186 to maintain reader engagement within the Results section.
Comments 2: Lines 101-107: Please briefly mention what equipment you used to analyze the blood biomarkers.
Response 2: Thank you for this helpful suggestion. We have removed the redundant text and rewritten Section 2.4 (lines 126–140) to integrate the description of the biomarker assays and the equipment used alongside the statistical methods.
Comments 3: Lines 118-140: Perhaps, instead of “blood-based biomarkers”, the title of this section could be “Statistical analysis and blood-based biomarkers” and integrate both sections into one.
Response 3: Thank you for the thoughtful recommendation. This comment aligns closely with the point raised in Comment 2, and we have revised the manuscript accordingly. Specifically, we rewrote Section 2.4 “Blood-based biomarkers” to clearly describe the sample matrices, assay methods, and analytical equipment used, removing previously duplicated content related to statistical analysis.
However, rather than merging Sections 2.4 and 2.5, we decided to retain separate sections for “Blood-based biomarkers” and “Statistical analysis” to preserve clarity and maintain a logical structure. In our view, keeping laboratory methodology and statistical procedures distinct allows readers to locate relevant information more efficiently and avoids overloading a single section.
We sincerely appreciate the reviewer’s insight, which helped us streamline and improve both sections.
Comments 4: Discussion section: Please discuss why some types of breeds or sex might have shown these results more frequently than others.
Response 4: Thank you for the valuable suggestion. We have added a summary of findings related to sex, neuter status, and breed distribution in the Discussion section (lines 334–337) to contextualize our results in relation to previous studies. Additionally, the issue regarding the 7-year age cutoff for defining senior dogs has been relocated to the Limitations section (lines 382–388) for better structural coherence and to highlight its relevance as a methodological consideration.
Reviewer 2 Report
Comments and Suggestions for Authors
General Comments: The authors examined whether four blood biomarkers were associated with stage of canine cognitive dysfunction syndrome (CDS), assessed in 77 dogs (14 of which were healthy controls) using three different behavioral questionnaires for CDS. They found that one biomarker, neurofilament light chain, showed the most consistent association with stage of CDS across all three questionnaires. Strengths of the study include the use of three different CDS questionnaires as well as four different biomarkers. Limitations include using biomarkers in blood rather than cerebrospinal fluid and owner-completed questionnaires (which, even though validated, have some well-known shortcomings), and failure to use imaging techniques to confirm brain changes. However, all of these limitations are stated in the Discussion (lines 367-378). I would add another limitation: the use of the 7-year cutoff for classifying dogs as seniors, given that most of the dogs in this study were small breeds, which tend to have longer lives than larger dogs. This is described in lines 318-324 of the Discussion, but it might fit better in the limitations section. My other comments and suggestions are detailed below.
Simple Summary: This is an excellent summary of your findings for nonscientists.
Abstract: As mentioned below in my comments on the Introduction, it would be good to explicitly state what is new about your study.
Keywords: The term “canine cognitive dysfunction syndrome” occurs in the title, so it will be picked up in searches. Consider replacing it with something like “behavioral questionnaire” to indicate that your study also included owner-completed questionnaires; this is important because your title does not mention behavioral assessments.
Introduction:
Lines 85-89: This paragraph could be stronger if you included specific hypotheses tested and predictions made based on existing studies whose findings are described in the Introduction. It is also important to explicitly state here (and in the Abstract) what is new about your study when compared with other studies on this topic (e.g., is this the first such study to use all three CDS questionnaires?).
Materials and Methods:
There are some places in this section where text is repeated:
Line 95: repeats number of dogs, which is stated in line 92.
Section 2.5 (begins line 131) largely repeats information in Section 2.4 Blood-based biomarkers. What is the difference between these two sections?
Section 2.3 Questionnaires (begins line 108): Please consider including the three questionnaires as Appendices so readers do not have to go to multiple other sources to fully understand your study.
Although dog breed is included in the results, it is never mentioned in Materials and Methods. For example, how was dog breed assessed? Did you rely on information in the owner-completed questionnaires, owner-supplied pedigrees or registration papers, or visual inspection? Note that visual inspection has been shown to be inaccurate:
Voith, V. L., Ingram, E., Mitsouras, K., & Irizarry, K. (2009). Comparison of Adoption Agency Breed Identification and DNA Breed Identification of Dogs. Journal of Applied Animal Welfare Science, 12(3): 253-262.
Voith, V. L., Trevejo, R., Dowling-Guyer, S., Chadik, C., Marder, A., Johnson, V., & Irizarry, K. (2013). Comparison of visual and DNA breed identification of dogs and inter-observer reliability, American Journal of Sociological Research, 3(2): 17-29.
Results:
Lines 172-174 repeats information stated several times previously. I would delete these lines and modify the sentence that follows to something like, “Based on classifications from the three CDS questionnaires, dogs were grouped accordingly…
Lines 292-294 repeats information stated several times previously - delete.
There are four tables and seven figures, which is a lot, but all seem necessary.
Discussion:
As mentioned above in General Comments, I would integrate lines 318-324 into the paragraph on study limitations.
Line 333: It would be helpful to readers if you would provide examples of confounding pathologies here.
Conclusions:
Line 387-388: I suggest adding “and behavioral assessments” after “blood biomarkers” to make the point that both can be helpful during veterinary visits.
Author Response
Comments 1: Lines 85-89: This paragraph could be stronger if you included specific hypotheses tested and predictions made based on existing studies whose findings are described in the Introduction. It is also important to explicitly state here (and in the Abstract) what is new about your study when compared with other studies on this topic (e.g., is this the first such study to use all three CDS questionnaires?).
Response 1: To emphasize the novelty of our study, we have updated the final paragraph of the Introduction to mention the application of all three validated behavioral questionnaires as well as the underlying hypothesis on biomarker changes (lines 90-96).
Comments 2: Keywords: The term “canine cognitive dysfunction syndrome” occurs in the title. Consider replacing it with something like “behavioral questionnaire.”
Response 2: Thank you for this thoughtful suggestion. Because our study relies on both behavioral questionnaires and blood biomarkers, we have added “behavioral questionnaire” to the keyword list (line 37). At the same time, we have retained “canine cognitive dysfunction syndrome” because it is the principal disease under investigation and a widely used search term in the field. We believe keeping both terms maximizes discoverability while accurately reflecting the scope of the work.
Comments 3: Repeated text in several places (e.g., line 95; lines 172–174; lines 292–294) and redundancy between Sections 2.4 and 2.5.
Response 3: We appreciate the reviewer for bringing this to our attention. All duplicate sentences at before line 95, before lines 172–174, and before lines 292–294 have been removed. In addition, the overlap between Sections 2.4 and 2.5 has been resolved: Section 2.4 has been fully rewritten (now lines 126–140) to describe the assay methods, and analytical equipment, whereas Section 2.5 now focuses solely on statistical procedures. These revisions eliminate redundancy and improve clarity.
Comments 4: Section 2.3 Questionnaires (begins line 108): Please consider including the three questionnaires as Appendices so readers do not have to go to multiple other sources to fully understand your study.
Response 4: I have included the full versions of the three behavioral questionnaires used in this study as appendices to ensure transparency and ease of reference for readers. These appendices are provided as supplementary files.
Comments 5: Although dog breed is included in the results, it is never mentioned in Materials and Methods. For example, how was dog breed assessed? Did you rely on information in the owner-completed questionnaires, owner-supplied pedigrees or registration papers, or visual inspection?
Response 5: Breed information is now detailed in lines 106-107. Specifically, breed was recorded from the owner-completed questionnaire and, when available, registration documents.
Comments 6: As mentioned above in General Comments, I would integrate lines 318-324 into the paragraph on study limitations.
Response 6: The discussion of the 7-year age threshold has been moved to the Limitations section (now lines 382–388), where it is presented together with other methodological constraints. We have also added a new reference ([24] Kim S.S, 2021) to support the rationale for alternative age cut-offs.
Comments 7: Line 333: It would be helpful to readers if you would provide examples of confounding pathologies here.
Response 7: Based on the referenced literature, we have added an example of a potential confounding factor to improve clarity and contextual relevance in the discussion. We appreciate the reviewer’s helpful comment.
Comments 8: Line 387-388: I suggest adding “and behavioral assessments” after “blood biomarkers” to make the point that both can be helpful during veterinary visits.
Response 8: We have incorporated the phrase “and behavioral assessments” after “blood biomarkers” in the Conclusions (now line 409) to emphasize that both modalities can support clinical decision-making. Thank you for highlighting this point.
Reviewer 3 Report
Comments and Suggestions for Authors
This is an interesting and timely study that addresses an important topic in veterinary neurobiology. The manuscript is overall well written and the experimental design carefully executed. The approach to investigating blood-based biomarkers of cognitive dysfunction in aging dogs is relevant, and the inclusion of behavioral questionnaires strengthens the translational value of the study.
The results are certainly intriguing and, in some cases, quite surprising. They contribute to the growing body of literature on canine cognitive dysfunction and suggest potential avenues for future clinical applications. However, several methodological and reporting issues require clarification to fully assess the reliability and interpretability of the findings.
While the study provides valuable insights into the potential use of blood-based biomarkers (Aβ40, Aβ42, NfL, GFAP) for assessing cognitive dysfunction in dogs, a major methodological limitation lies in the sampling protocol—specifically, the decision to perform a single blood extraction per subject. This cross-sectional design restricts the capacity to evaluate intraindividual variability and temporal dynamics of biomarker expression.
Studies in both human and veterinary neurodegeneration have demonstrated that levels of biomarkers can be affected by transient physiological variables, including stress, circadian rhythms, and physical activity. A single measurement therefore introduces the risk of misinterpreting transient fluctuations as trait-level differences between cognitively impaired and control groups.
Furthermore, cognitive dysfunction is, by nature, a progressive and fluctuating condition. Without repeated sampling over time, it is not possible to determine whether the biomarkers assessed are stable indicators of disease state, or if they merely reflect episodic or context-dependent changes. This is particularly relevant when attempting to correlate molecular findings with behavioral data obtained from owner-based questionnaires, which are themselves subject to a degree of subjectivity and temporal variability.
To strengthen the conclusions and clinical applicability of these findings, future studies should incorporate longitudinal sampling protocols, ideally involving multiple timepoints across disease progression. This would not only improve the robustness of biomarker validation but also help to delineate individual trajectories of cognitive decline and their molecular correlates.
One important limitation of the study lies in the insufficient methodological detail regarding biomarker quantification. Although the authors state that concentrations of Aβ40, Aβ42, NfL, and GFAP were measured, they do not clarify which biomarkers were analyzed in plasma and which in serum, despite collecting both fractions. This omission is problematic, as matrix-specific differences can significantly affect biomarker levels and interpretation.
Moreover, no information is provided on the analytical procedures or detection methods used to quantify these biomarkers. There is no mention of the use of commercial kits, immunoassay platforms (e.g., ELISA, SIMOA), or previously validated protocols. Surprisingly, only in Abstract and Discussion ELISA is specified as the technique employed. Why is it not described in the Methods section? It would have been appropriate to include brief methodological descriptions or references to earlier studies (e.g., “Serum NfL concentrations were measured as described in…”) to ensure transparency and allow reproducibility.
The authors should clearly specify:
- The biological matrix used for each biomarker,
- The quantification technique or platform employed,
- Whether methods were validated or referenced from previous research.
Author Response
Comments 1: To strengthen the conclusions and clinical applicability of these findings, future studies should incorporate longitudinal sampling protocols, ideally involving multiple timepoints across disease progression. This would not only improve the robustness of biomarker validation but also help to delineate individual trajectories of cognitive decline and their molecular correlations.
Response 1: We appreciate this insightful comment. We concur that a single blood draw may be insufficient given the progressive and potentially fluctuating nature of CDS. In future studies, we plan to implement longitudinal sampling at multiple timepoints across disease progression to enhance the robustness of biomarker validation and to better capture individual trajectories of cognitive decline.
Comments 2: One important limitation of the study lies in the insufficient methodological detail regarding biomarker quantification. Although the authors state that concentrations of Aβ40, Aβ42, NfL, and GFAP were measured, they do not clarify which biomarkers were analyzed in plasma and which in serum, despite collecting both fractions. This omission is problematic, as matrix-specific differences can significantly affect biomarker levels and interpretation.
Response 2: We fully agree with the reviewer’s observation. During manuscript preparation, Sections 2.4 and 2.5 were inadvertently duplicated, leaving key methodological details unclear. We have now deleted the redundant text and completely rewritten Section 2.4 (lines 126–140) to specify the biological matrix and analytical platform for each biomarker. The revised section also notes that all assay procedures follow the validated protocols described in references [13]–[22]. These changes clarify matrix-specific information, cite the methodological groundwork, and enhance reproducibility. Thank you again for this valuable suggestion.
Round 2
Reviewer 3 Report
Comments and Suggestions for Authors
Thank you for your response. While I appreciate your acknowledgment of the limitations related to single timepoint sampling and your intention to address this in future studies, I strongly recommend that this issue be clearly stated as a limitation in the Discussion section of the current manuscript.
Given the progressive and fluctuating nature of CDS, relying on a single blood draw significantly impacts the generalizability and interpretability of the findings. Explicitly recognizing this limitation would help contextualize the conclusions.
Author Response
Comments 1: I strongly recommend that this issue be clearly stated as a limitation in the Discussion section of the current manuscript. Given the progressive and fluctuating nature of CDS, relying on a single blood draw significantly impacts the generalizability and interpretability of the findings. Explicitly recognizing this limitation would help contextualize the conclusions.
Response 1: We appreciate the reviewer’s insightful recommendation and fully agree that the reliance on a single blood draw should be overtly discussed as a limitation. Accordingly, we have added the following sentence to the Discussion (lines 396-399)
By explicitly acknowledging this constraint and outlining our plan for multi-timepoint sampling in subsequent work, we believe the manuscript now provides clearer context for the study’s conclusions. Thank you for helping us improve the rigor and transparency of our Discussion section.